



# Empirical correction of systematic orthorectification error in Sentinel-2 velocity fields for Greenlandic outlet glaciers

Thomas R. Chudley[1], Ian M. Howat[1,2], Bidhyananda Yadav[1], Myoung-Jong Noh[1]

[1]Byrd Polar and Climate Research Center, Columbus, OH, USA
[2]School of Earth Sciences, Ohio State University, Columbus, OH, USA

*Correspondence to*: Tom Chudley (chudley.1@osu.edu)

**Abstract.** By utilising imagery from overlapping orbits, the Sentinel-2 programme offers high-frequency observations of high-latitude environments well in excess of its 5-day repeat rate, which is valuable for obtaining large-scale records of rapid environmental change. However, the production of glacier velocity datasets from optical feature tracking of Sentinel-2 imagery

is limited by the orthorectification error in ESA products, which introduces significant systematic errors (on the order of tens of metres) into displacement fields produced from cross-track image pairs. Here, we use temporally complete glacier velocity datasets to empirically reconstruct systematic error, allowing for the corrected velocity datasets to be produced for four key fast-flowing marine-terminating outlets across the Greenland Ice Sheet between 2017 – 2021. We show that corrected data agrees well with comparison velocity datasets derived from optical (Landsat 8) and synthetic aperture radar (Sentinel-1) data.

The density of available velocity pairs produces a noisier dataset than for these comparative records, but a best-fit velocity reconstructed by time-series modelling can identify periods of rapid change (e.g., summer slowdowns), even where gaps exist in other datasets. We use the empirical error maps to identify that the commercial DEM used to orthorectify Sentinel-2 scenes over Greenland between 2017 – 2021 likely shares data sources with freely available public DEMs, opening avenues for the analytical correction of Sentinel-2 glacier velocity fields in the future.

## 1 Introduction

Continuous glacier velocity datasets derived from medium-resolution satellite programmes have become increasingly available in recent years, forming a key part of investigations into ice discharge (King *et al.* 2018; Gardner *et al.* 2018; Mankoff *et al.* 2020), glacier dynamics (Poinar and Andrews, 2021; Dehecq *et al.* 2019), and characterisation of seasonal glacier behaviour (Vijay *et al.* 2021; Moon *et al.* 2014). Globally comprehensive scene-pair velocity fields from medium-resolution satellite data

are available from both optical feature-tracking and SAR speckle-tracking techniques: e.g. for Landsat 8 optical data, the ITS_LIVE programme (Gardner et al., 2018, 2021); and for Sentinel-1 SAR data, the MEaSUREs and PROMIICE programmes for Greenland (Joughin, 2021a, Solgaard *et al.* 2021) and the RETREAT programme for glaciers and ice caps (Friedl *et al.* 2021). The Sentinel-2 mission holds further promise in deriving glacier velocities from optical imagery compared to Landsat 8, offering an improved repeat time of 5 days (with both Sentinel-2a and -2b) compared to 16 days, and a resolution



of 10 m in the visible and near-infrared portion of the electromagnetic spectrum compared to 30 m (15 m panchromatic) for Landsat 8. This dense temporal coverage increases the chances of finding cloud-free image pairs, particularly when making use of cross-track imagery at high latitudes. However, as yet, the use of Sentinel-2 velocity fields - particularly in the form of large-scale public datasets - are limited.

A particular problem for Sentinel-2 feature tracking is the presence of systematic orthorectification errors. These are lateral

off-nadir offsets in orthorectified satellite imagery resulting from vertical differences between the Digital Elevation Model (DEM) surface used to orthorectify the imagery and the true surface at the time of acquisition. Over solid bedrock, these offsets occur due to DEM errors, but the issue is exacerbated in glacial environments, where significant (10s of metres or more) real elevation change may occur between the DEM and image acquisition times due to changes in ice surface elevation (hereafter $\Delta h$) resulting from, mainly, sustained flow acceleration, increased surface melt and, subsequently, rapid ice thinning (e.g. King

et al. 2020). When tracking displacement between two optical scenes from the same orbital path (in Sentinel mission terminology, the 'relative orbit', hereafter RO), orthorectification errors will be the same across the two images and will be eliminated in the final displacement map. However, in scene pairs from different orbits, a systematic error will be present as the vector sum of the two orthorectification errors. Sentinel-2 is particularly vulnerable to orthorectification error, suffering from an order-of-magnitude greater terrain bias than Landsat 8 (Altena and Kääb, 2017). This is in part due to the wide viewing

angle of Sentinel-2 compared to Landsat 8: with Sentinel-2's swath width of 290 km, a vertical DEM error of $\Delta h$ can result in a worst-case offset of $\sim\Delta h/5.4$ at the maximum off-nadir distances, whilst for Landsat 8's swath width of 185 km this is only $\sim\Delta h/7.8$ (Kääb et al. 2016). However, in the L1C and L2A data provided by ESA, the large errors are also related to the DEM chosen to orthorectify Sentinel-2 imagery. Until the 23rd August 2021 (30th March 2021 for Europe and Africa), the commercial PlanetDEM 90 m global elevation model ([https://planetobserver.com/global-elevation-data/](https://planetobserver.com/global-elevation-data/)) was used to orthorectify Sentinel-

2 data. Little public information exists as to which data sources were used to construct the PlanetDEM outside of the Shuttle Radar Topography Mission (SRTM) acquisition zone. However, Kääb et al. (2016) suggest that high-latitude source datasets are shared with the Viewfinder Panorama 3" DEM (de Ferranti, 2014), which use, among other sources, 20th century topographic maps to reconstruct high-latitude ice topography (J. de Ferranti, pers. comm.). This decades-old source data could explain the significant vertical DEM errors compared to Landsat 8, which, in Collection-2 processing, uses more recent region-

specific elevation models at high latitudes (Franks et al. 2020), such as the ArcticDEM, Greenland Ice Mapping Project (GIMP) DEM, and Alaskan National Elevation Dataset. As a result, orthorectification errors remain a significant issue for producing consistent Sentinel-2 glacier velocity fields. Kääb et al. (2016) recommend that feature tracking using cross-track image pairs should be performed only for ice displacements that are at least one order of magnitude larger than the expected orthorectification error, whilst Nagy et al. (2019) recommend not using cross-track pairs at all. This limits the benefits of

Sentinel-2's dense data coverage, reducing the number of available image pairs to only those with temporal baselines of 5 (10, 15 etc.) days. Being able to remove or account for the off-nadir orthorectification error is highly desirable to unlock the full potential of dense Sentinel-2 temporal coverage at high latitudes.





A range of solutions have been implemented to account for orthorectification errors in PlanetDEM-corrected Sentinel-2 imagery. With access to the PlanetDEM 90, Ressl and Pfeifer (2018) were able to generate a predicted offset field over Austria

by using the PlanetDEM, the known orbits of Sentinel-2, and reference DEM (the AustriaDEM) to provide a 'truth' reference surface. Here, rays were projected from the satellite orbital path to the reference DEM and intersected with the PlanetDEM to derive the off-nadir offset. However, this method not only requires access to the PlanetDEM (which is not freely available) but also a reference elevation model that is accurate at the time of image acquisition, which is challenging for glaciated regions where, in places, surface elevations are changing significantly on an interannual timescale. Altena and Kääb (2017) present an

alternative method for glacier velocity fields that does not require any prior knowledge of $\Delta h$. If glacier flow is known a priori, it is possible to map the offset onto this flow direction as the offset vector always occurs perpendicular to the flight path (or, for dual orbits, along the epipolar line between the two satellite locations). This is an elegant solution that can be implemented in normal image matching pipelines without requiring elevation data, but comes with two primary limitations. The first is the assumption that flow direction is stable over time, which is justified on sub-decadal timescales for ice streams and glaciers that

are not undergoing significant changes in their geometries, such as occurs during surges. However, the second limitation of this method is that comprehensive coverage is restricted by two criteria: (i) when the flow direction bearing is in the same direction as the epipolar line, the displacement will be mapped to infinity; and (ii) when displacement is within the measurement error, the same effect can occur. As a result, the authors filter velocities where the flow direction is within 20° of the epipolar line, and where displacement is >2.5 times the matching accuracy. Hence, the final corrected velocity field is

discontinuous. To produce continuous and dense velocity fields taking advantage of the entire Sentinel-2 record, it would be desirable to develop a method that does not require any prior knowledge of $\Delta h$ whilst still being able to produce a geographically complete record of velocity.

Here, we take advantage of five years of Sentinel-2 imagery to generate empirical corrections for systematic orthorectification error in ice surface velocity fields at four key marine-terminating outlet glaciers around the Greenland Ice Sheet. We describe

the process by which we produce dense and continuous velocity datasets from 2017 – 2021, before validating our results by comparing them to publicly available velocity datasets at four key outlet glaciers.

## 2 Methods

We produce and present velocity data for four major marine-terminating glaciers: Sermeq Kujalleq (Store Glacier), Sermeq Kujalleq (Jakobshavn Isbræ), Helheim Glacier, and Kangerlussuaq Glacier (fig. 1). As two of these glaciers share a

Greenlandic name, we hereafter refer to them by their alternative names used in scientific literature (Store Glacier and Jakobshavn Isbræ).



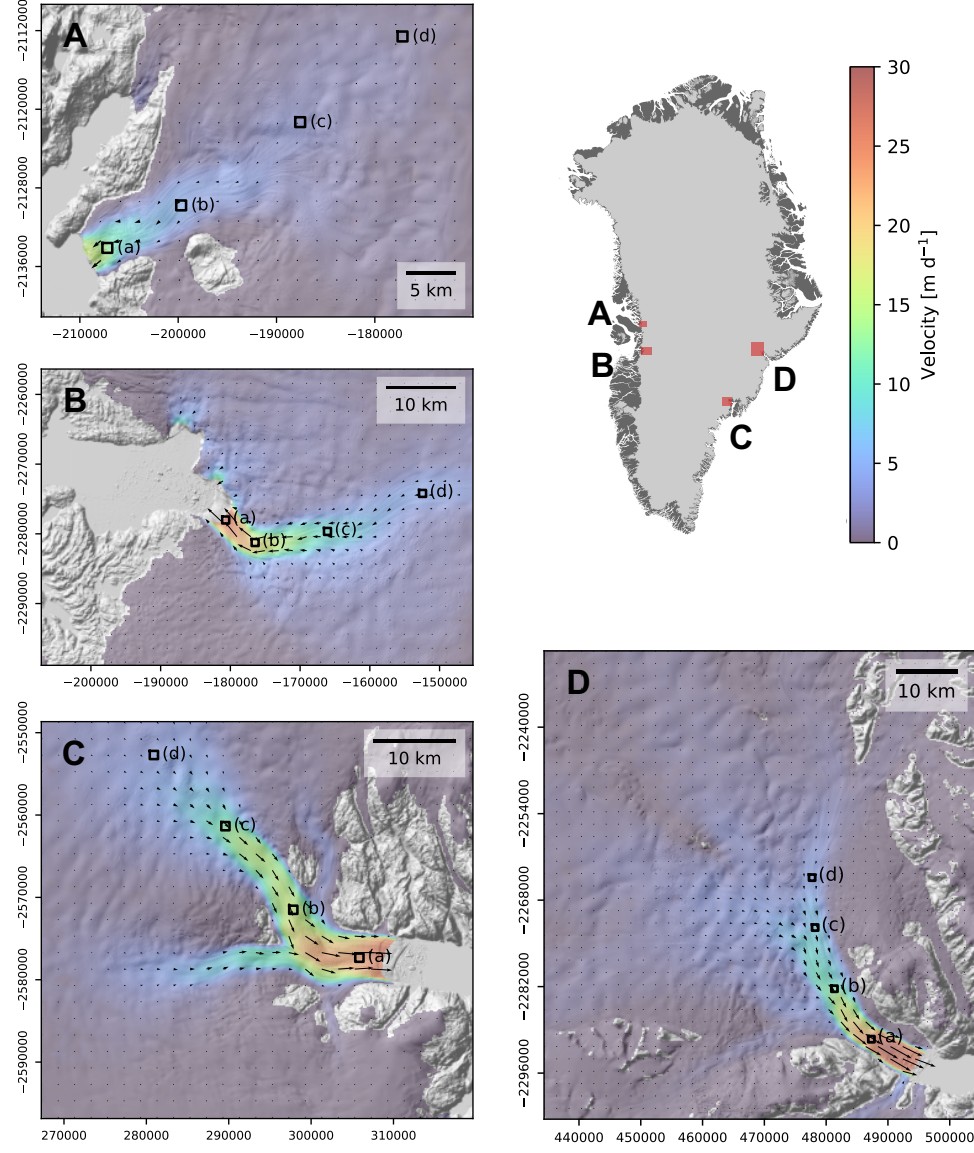

**Figure 1: Reference velocity fields (median velocity of repeat-orbit pairs; see section 2.1.2) for the four marine-terminating glaciers presented in this paper, ordered anti-clockwise from North: (a) Store Glacier; (b) Jakobshavn Isbræ; (c) Helheim Glacier; and (d) Kangerlussuaq Glacier. 1 × 1 km sample sites a-d are marked for each glacier - see Table S2 for precise coordinates. Backgrounds are GIMP DEM hillshades (Howat, 2017); coordinates in NSIDC Polar Stereographic North. Inset: location in Greenland of all outlet glacier AOIs (red).**





**Figure 2: Workflow for deriving corrected and filtered glacier velocity fields from Sentinel-2 Level-2A data.**



As orthorectification error is a function of satellite geometry, it should result in a consistent offset (in units of absolute displacement rather than velocity) across all velocity fields generated from the same RO pairs. We assume that other errors in the velocity field (e.g. image matching error, coregistration error) are (a) random; and (b) do not correlate with specific RO pairings. Hence, by drawing from a large base dataset, we can infer the average orthorectification error over the study period

for specific RO pairs by measuring the average offset between (i) the ice displacement measured from Sentinel-2 scenes and (ii) the expected displacement from a reference velocity field. We refer to this difference as the orbit-pair offset field, and subtract it from measured fields to generate final corrected fields (fig. 2).

## 2.1 Data production

### 2.1.1 Velocity field production

Data are produced at four marine-terminating glaciers, matching the spatial extent of the areas of interest (AOIs) in the MEaSUREs Selected Glacier Site Velocity Maps from Optical Images collection (Howat, 2020). We downloaded all Bottom-Of-Atmosphere corrected Level-2A (L2A) Sentinel-2 data with a cloud cover < 50% from the Amazon Web Services (AWS) Registry of Open Data using the `sat-search` STAC API ([https://github.com/sat-utils/sat-search](https://github.com/sat-utils/sat-search)). Sentinel-2 data is staged using a pre-defined tiling scheme based on the Military Grid Reference System (MGRS). The pre-processing pipeline includes:

subsetting Sentinel-2 tiles for each glacier bounding polygon AOI; reprojecting the raster to a common WGS 84 / National Snow and Ice Data Centre (NSIDC) Sea Ice Polar Stereographic North (EPSG:3413) projected coordinate system; mosaicking the adjacent overlapping tiles; and finally clipping the mosaicked raster to the glacier Area of Interest (AOI). We only considered scenes until the 23rd August 2021, when the L1C orthorectification process switched to a new geolocation procedure and underlying DEM (section 4.3).

Velocity fields at 100 m resolution were produced using feature tracking methods, performed using a Directional Weighted Filtering (DWF) algorithm based on the Surface Extraction from TIN-based Search-space Minimization (SETSM) approach (Noh and Howat, 2019). The method was originally developed for precisely estimating the surface displacement map (SDM) by compensating relative sensor model biases (minimising co-registration errors) and removing orthorectification errors caused by height changes through true DEMs. For the purpose of this research, we modified the algorithm to use orthorectified images

directly, and applied the relative sensor model bias compensation module as co-registrations. The SDM processing is fully automated except in using an a priori, or seed, velocity field to specify maximum displacements for determining the initial resolution in the coarse-to-fine processing scheme (Noh and Howat, 2019). Here, we used InSAR-derived velocity fields between 2016 and 2017 (Joughin, 2021b) as the seed.

### 2.1.2 Estimating orbit-pair offsets

The ROs of scenes are obtained from the scene metadata. Pairs of images acquired from the same RO are hereafter referred to as *repeat-track pairs*, and those from different ROs *cross-track pairs*. For any given outlet glacier, certain combinations of



orbit pairs may have anywhere from a few to >100 velocity fields. Where there are fewer than five velocity fields over the study period (2017-2021) for a given orbit pair, an empirical offset field is not constructed and velocity fields are not processed further.

A reference flow field is constructed using the median $U$ and $V$ velocity values (in $x$ and $y$ EPSG:3413 Polar Stereographic North grid directions) from all (2017-2021) repeat-track velocity fields. Before processing, uncorrected velocity fields are filtered using a $3 \times 3$ median filter to reduce noise. For individual velocity fields, the expected displacement is calculated from these reference flow fields and the temporal baseline of the scene pair. The offset between the uncorrected displacement and the expected displacement is then calculated. Empirical orbit-pair offset fields are generated as the median offset for each orbit

pair.

We note that over the course of the study period, ongoing glacier surface elevation change will continue to change $\Delta h$ and hence the orthorectification error will not be constant. However, our method of estimating orthorectification error across the entire 2017-2021 study period implicitly assumes a constant $\Delta h$. We could improve this assumption by assessing offsets on shorter timescales – such as annually – but shorter timescales (smaller sample sizes) result in a notable reduction in the number

of cross-track pairs available for correction (i.e. satisfying our threshold of 5 available velocity fields), and a lower quality of offset fields even for cross-track pairs where sufficient data is available. However, given the large initial $\Delta h$ values, surface elevation change over the study period likely has a negligible impact on the orthorectification error. Using surface elevation change values from Smith *et al.* (2020) as approximations, maximum surface elevation change rates at our study sites range between -0.3 m a$^{-1}$ (Store Glacier) to -3.6 m a$^{-1}$ (Kangerlussuaq Glacier). Maximum estimated $|\Delta h|$ values (Section 3.1), which

roughly correlate with these values, range between ~160 and ~400 m. Over the four-year study period, this results in a potential time-dependent error in $\Delta h$ of between 0.3 and 2.2% at our study glaciers. Applying these uncertainties to typical maximum offsets (directly correlated with $\Delta h$) of between 40 and 60 metres shows that, even using worst-case assumptions, offset vector errors range between ±0.2 and ±1.3 m, values which are subsumed by other error terms (e.g. miscorrelation and coregistration).

### 2.1.3 Velocity correction and filtering

Once orbit-pair offsets have been constructed, uncorrected velocity fields are converted to absolute displacement, corrected using the appropriate orbit-pair displacement offset field, and converted back to velocity. Displacements are corrected only over ice as defined in the Greenland Ice Mapping Project (GIMP) ice mask (Howat *et al.* 2014; Howat, 2017): vertical DEM error outside of this mask is assumed to come only from elevation measurement error and not from surface elevation change. Due to changing ice boundaries at marine-terminating locations, areas within the GIMP ocean mask are not filtered or removed,

but ice velocities beyond the extent of the GIMP ice mask should not be considered reliable.

To remove erroneous velocity measurements, areas within the GIMP ice mask are filtered where flow directions are >20° offset from the reference flow field. If, after filtering, no data remains (<1% of the ice area has valid velocity measurements) the field is discarded and no output data is generated.



### 2.1.4 Error assessment

A first-order estimate of error is taken as the root mean square error (RMSE) of the absolute velocity of the bedrock area (as defined by the GIMP bedrock mask). RMSEs tended to be low, with the median RMSE consistently beneath <0.5 m d$^{-1}$ for the study glaciers discussed in this paper (fig. S1). Additionally, the mean and standard deviation of the $U$ and $V$ velocity fields of the bedrock area are also recorded, in order to assess systematic error within individual flow fields due to e.g. poor co-registration. Where the mean of the $U$ or $V$ velocity is greater than one standard deviation away from zero, the field is
considered to have a systematic error and is not included for presentation in this study (section 2.2).

### 2.2 Data presentation

### 2.2.1 Sampling

To present time series' of glacier surface velocity, we sample four sites of increasing distance from the calving front at each of our sample glaciers (Figure 1; Table S1). We sample across a 1 × 1 km area, calculating the median velocity and error across
this sample region. We filter out data points where < 70% of the sample region contains data, or where the error is > 5 m d$^{-1}$. We further filter fields where the temporal baseline is only two days, where errors were significantly greater than any other baselines (Figure S1).

### 2.2.2 Gaussian process regression

The output from our velocity correction process produces a dense time-series of varying error estimates. Hence, we use
Gaussian process (GP) regression (Rasmussen and Williams, 2006) to estimate true velocity from our sampled time-series observations. Two particular properties of GP regression make it useful for the current application: (i) the Bayesian nature of the method accommodates the incomplete velocity record, producing a smooth, nonlinear, interpolated output; and (ii) the probabilistic model can incorporate uncertainty estimates and provides an empirical confidence interval to predictions.

We performed GP regression using the `GaussianProcessRegressor` implementation in the Python `scikit-learn` library. We model ice velocity as the sum of two kernels (covariance functions), with the mean function equal to the mean ice velocity of the sample dataset. Following Rasmussen and Williams (2006), we implement seasonal variability and short-medium term variability kernels (ignoring long-term variability over the five-year period we assess). Seasonal variability is incorporated as an exponential sine squared kernel with a fixed periodicity of one year and the length-scale (controlling its
smoothness) as a free parameter. To allow the period to vary over length-scales, the product with a radial basis function kernel with a free length-scale is taken. We implement short-medium term variability as a rational quadratic kernel, with free length-scale and alpha (controlling the diffusivity of the length-scale) parameters. Finally, we incorporate velocity error estimates (section 2.1.3) into modelling directly via the `alpha` parameter of the `GaussianProcessRegressor`.





### 2.3 Supplementary data

As orthorectification errors scale with *Δh*, we compare generated offset maps to DEM difference as a proxy for *Δh*. The first DEM is the *Viewfinder Panorama 3" DEM for Greenland (Version 1)* (de Ferranti, 2014; hereafter the 'Viewfinder DEM'). Kääb *et al.* (2016) inferred that the Viewfinder DEMs likely shared source data with the PlanetDEM for a test site in Northern Norway. The second DEM is the *GIMP DEM from GeoEye and WorldView Imagery (Version 1)* (Howat *et al.* 2014; 2017; hereafter the 'GIMP DEM'), which was produced from imagery between 2009 – 2015.

We compare our generated velocity fields to two other public datasets. To compare with medium-resolution SAR velocity fields, we make use of MEaSUREs *Greenland 6 and 12 day Ice Sheet Velocity Mosaics from SAR (Version 1)* velocity fields from speckle-tracked Sentinel-1 data (Joughin *et al.* 2018; Joughin, 2021a). These were downloaded from the NSIDC data portal. To compare with optical velocity fields derived from Landsat 8 imagery, we make use of the ITS_LIVE dataset (Gardner *et al.* 2018, 2021). These were downloaded using the ITS_LIVE API, and, to match our Sentinel-2 data thresholds, filtered to

velocity fields where (i) the maximum interval between data pairs was 30 days; and (ii) at least 1% of the data contained valid pixels. For both Sentinel-1 and Landsat 8 fields, we extracted time-series data in the same way as for Sentinel-2 data: i.e. filtering to data that covers at least 70% of the 1 × 1 km sample region, and < 5 m d⁻¹ error.

### 3 Results

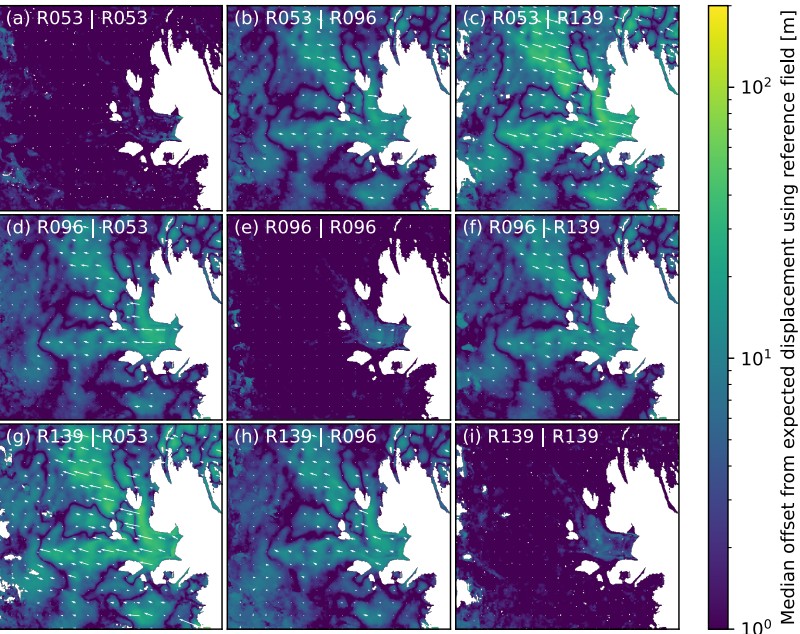

**Figure 3: Median offset between measured displacement and expected displacement (from reference velocity field) for different orbital pairs at Helheim Glacier. Magnitude is shown in colour; vectors are shown as white arrows.**

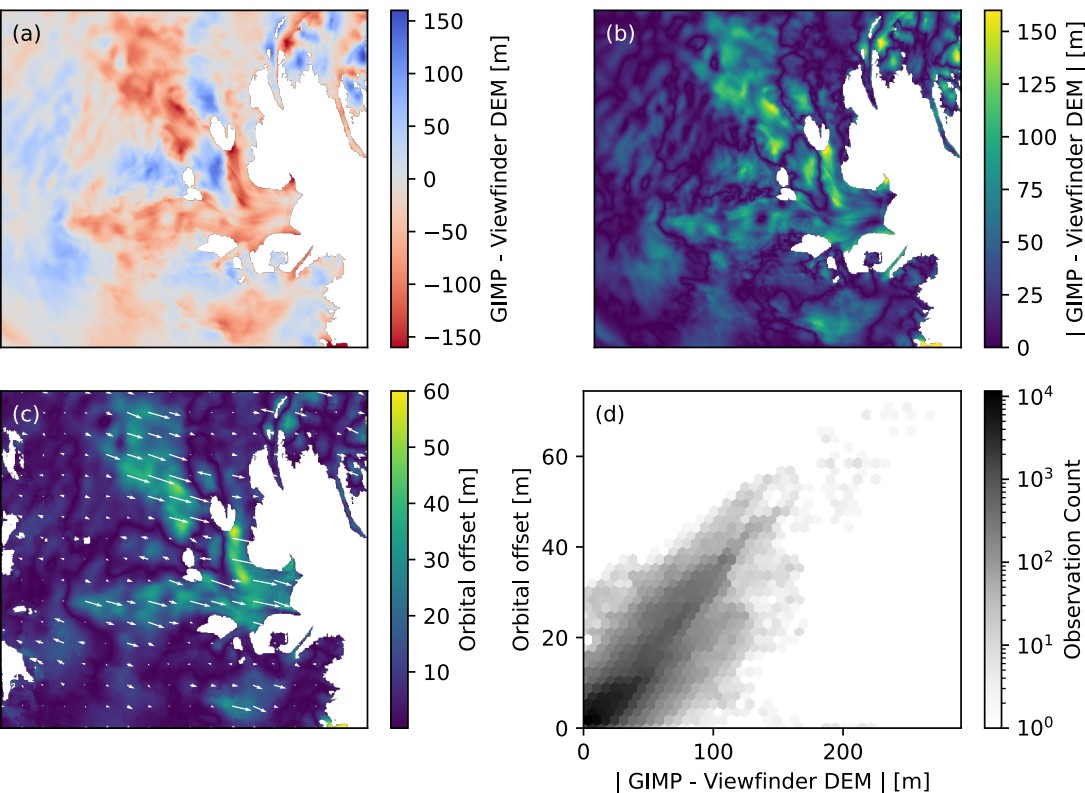

**Figure 4: Comparison between DEM vertical differences and velocity offset for Helheim Glacier. (a) DEM difference between the Viewfinder DEM and the GIMP DEM. Areas where the GIMP DEM has a lower elevation value are in red, and a higher elevation value in blue. (b) Absolute vertical difference between the Viewfinder DEM and GIMP DEM. (c) Median offset between the measured displacement and expected displacement for velocity fields taken from ROs 053 and 139 (identical to fig. 3c). (d) Hexbin density plot comparing the absolute vertical differences between the Viewfind DEM and GIMP DEM with the empirically determined orbital displacement.**

### 3.1 Orbital offset generation

Empirically determined systematic offsets can reach values on the order of tens of metres at outlet glaciers around Greenland. Comparing offset fields constructed for various orbit pairs at Helheim Glacier (fig. 3) shows that the behaviour of empirical offset fields are consistent with theory. Offset fields for repeat orbit pairs (fig. 3a, e, i) have negligible offset, whilst the offset is largest for the cross-track pairs from ROs 053 and 139 (fig 3c, g), which have the greatest distance between their respective orbital paths (fig. S2). Empirical offset vectors are, to within reasonable error, uniformly parallel and occur along the epipolar line of the satellite pair viewing geometry, consistent with the hypothesis that these systematic errors occur due orthorectification error in the off-nadir direction.

If the measured offsets are due to orthorectification error, they should hypothetically directly correlate to the $\Delta h$ between the times of DEM and image acquisition. We can thus validate that our offsets are meaningful by comparing them to the DEM





difference between the Viewfinder DEM, which we use as a proxy for the PlanetDEM (section 2.3), and version 2 of the GIMP DEM, which acts as our 'true' elevation, or at least a closer approximation to the surface elevation at the time of image acquisition (Fig. 4). The spatial pattern of $|\Delta h|$ (fig. 4b) matches the magnitude of the orbital offset (fig. 4c), with a strong, positive correlation ($R^2 = 0.67$, $p < 0.01$) between the two values (fig. 4d). The direction of the offset (white vectors in fig. 4c) is also predicted by the direction of $\Delta h$ (fig. 4a): where $\Delta h$ values are negative, offset error occurs in the ESE direction (towards
RO 053), whereas positive $\Delta h$ values occur where offset errors are in the WNW direction (towards RO 139).

The efficacy of the orbital correction fields is visualised for an example case at Jakobshavn Isbræ (fig. 5), which shows the relative ability of corrected and uncorrected glacier velocity fields to properly capture the magnitude and extent of a summer slowdown occurring in late July 2019 (see also figs. 6 and 7 for time series of this event). In the uncorrected velocity fields (fig. 5a and b), orthorectification error in the cross-track velocity fields (ROs 025 / 111 and ROs 068 / 025 respectively)
introduce an apparent difference in the final velocity field in excess of 10 m d$^{-1}$ at the calving front and 2 m d$^{-1}$ even tens of kilometres inland, a rate of change that is unphysical. After correction (figs. 5d and e), this change is reduced to ~2-3 m d$^{-1}$ at the front and negligible amounts inland (fig. 5f), an observation that is in line with contemporaneous Sentinel-1 observations (fig. 5g-i).

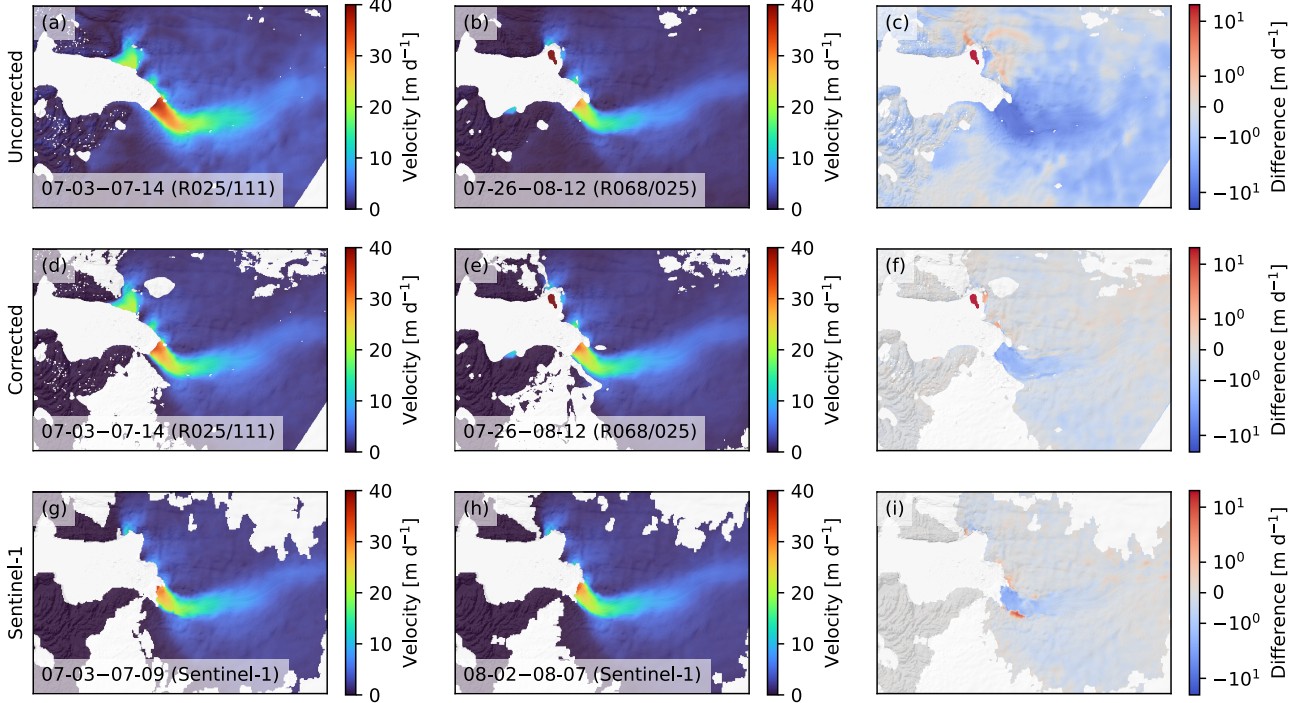

**Figure 5: Velocity difference maps at Jakobshavn over a period of summer slowdown in 2019 (see figs. 6 and 7). (a) Velocity field pre-slowdown, from uncorrected Sentinel-2 feature tracking. (b) Velocity field post-slowdown, from uncorrected Sentinel-2 feature tracking. (c) Difference between pre- and post-slowdown from uncorrected Sentinel-2 feature tracking. (d-f) Same as (a-c), but for corrected Sentinel-2 feature tracking. (g-i) Same as (a-c), but for contemporaneous Sentinel-1 speckle tracking. Note log scale used on difference maps. Backgrounds are GIMP DEM hillshades (Howat, 2017).**



**Figure 6: Corrected (blue) and uncorrected (red) velocity time-series for four sample glaciers in 2019. Top to bottom are points a-d in fig. 1. Black line and diamonds mark the MEaSUREs Sentinel-1 velocity dataset for comparison.**

### 3.2 Ice velocity time series

For the sampled 1 × 1 km sectors of our four study glaciers, we compare our corrected Sentinel-2 data against other data sources. Comparing our corrected velocity data to the uncorrected dataset shows the improvement that our empirical correction has (fig. 6). Uncorrected data shows a characteristic error distribution, with points offset from the Sentinel-1 -derived velocities based on their constituent orbit pairs and their temporal baseline. Short-baseline pairs are increasingly offset from the reference dataset value, even where orbit pairs are identical (fig. S3). This is because orthorectification error is an absolute displacement offset, and as such becomes a higher relative error component of short-baseline velocities. In all cases, the corrected velocity data converges near the reference Sentinel-1 time-series. The correction is greatest for observations at Jakobshavn Isbræ and Kangerlussuaq Glacier, where dynamic thinning and thus $\Delta h$ are the largest.



**Figure 6 (cont).**

The high density of data points, in combination with the relatively high uncertainty of optical feature tracking, results in a
noisy dataset compared to the SAR-derived data. The use of GP regression highlights an effective way of interpolating a 'true'
continuous time-series from this dataset (blue line in fig. 7), whilst accounting for sample-specific error values and time-
variable data densities. Across all sites, the median difference between the Sentinel-1 record and the GP fit is 0.08 m d$^{-1}$; 68%
(95%) of values lie within 0.41 (0.73) m d$^{-1}$ of one another; and 90.5% of values lie within their errors (Table S2). These
differences, on the order of decimetres, align with our error estimates from off-ice displacement of between 0.3 and 0.5 m d$^{-1}$
on average (fig. S1). The lowest level of agreement between the two datasets occurs at Helheim Glacier point b, where only
52.3% of Sentinel-1 velocity values lie within error of the Sentinel-2 GP fit, and the Sentinel-1 record is, on average, 0.47 m
d$^{-1}$ slower than that of the GP fit.

**Figure 7: Corrected Sentinel-2 (blue dots), ITS_LIVE Landsat-8 (orange triangles), and MEaSUREs Sentinel-1 (black diamonds)**
**velocity time-series for four sample glaciers in 2019. Top to bottom are points a-d in fig. 1. Blue line marks the output of the Gaussian process regression, with the blue shading marking the 2-sigma uncertainty bound.**

This high level of agreement also occurs in comparison with the ITS_LIVE dataset derived from optical Landsat 8 imagery (orange triangles in fig. 7). Across all sites, the median difference between the ITS_LIVE record and the GP fit is 0.12 m d$^{-1}$; 68% (95%) of values lie within 0.29 (0.63) m d$^{-1}$ of one another; and 87.0% of values lie within error of each other (Table S3).

The increased density of the Sentinel-2 record relative to the Landsat 8 record allows for a higher temporal precision in identifying rapid drainage events. For instance, the summer slowdown at sites a and b at Store Glacier is often not well captured in the Landsat 8 record, where a sparse record means that the precise nature and timing of the slowdown cannot be ascertained, whilst the dense Sentinel-2 dataset and the GP fit captures the asymptotic slowdown well (fig. S4). The Sentinel-2 dataset is also of sufficient precision to be able to assess changing strain rates across the same slowdown (fig. S5). However, the Landsat



8 record performs better at lower velocity values (sub-4 m d$^{-1}$ – e.g. site d at Store Glacier), likely because any errors in the empirically-derived displacement field have greater relative influence at low flow velocities.

**Figure 7 (cont).**

## 4 Discussion

### 4.1 Data quality

In the absence of empirical correction, orthorectification errors in glacier velocity can result in errors in excess of 10 m d$^{-1}$ in short-baseline velocity fields at Greenlandic outlet glaciers (figs. 5, 6). As orthorectification error is linearly proportional to $\Delta h$ between the orthorectification DEM and the true glacier surface (fig. 4), errors are greatest at glaciers where ice surface elevation has changed the most. This is exemplified by Store Glacier, which is notable for its long-term stability induced by a



sill at the glacier terminus (e.g. Morlighem *et al.* 2016), and has relatively low orthorectification errors prior to correction (fig. 6).

The empirical corrections described in this paper demonstrably reduce the influence of systematic orthorectification error on Sentinel-2-derived glacier velocity fields, reducing the variance from the order of metres to decimetres (figs. 6, 7). The temporal density of observations means that short term variations in speed, such as the slowdown at the terminus of Store

Glacier, can be well resolved. The noise of the dataset highlights the utility of effective filtering and time series modelling, such as GP regression (fig. 7), in extracting a continuous velocity estimate – following other studies that have made use of, for example, Kalman filtering (King *et al.* 2018) to similar effect – and provide the advantage of a greater accuracy through the synthesis of multiple estimates. GP fits over the four glaciers assessed here differ from Sentinel-1 estimated by, on average, only 0.08 m d$^{-1}$, and 90.5% of estimates lie within error (table S2). Unlike monthly averaging, the use of GP regression retains

the ability to capture short-term variation at ~weekly timescales and provides a time-varying confidence interval based on training data uncertainty and coverage.

The data display high agreement with comparative datasets sourced from Sentinel-1 and Landsat 8 (fig. 7). Although Sentinel-1 has been presented as a 'true' reference velocity in this study, it is of note that when only two of the three records presented agree with each other, which two will vary: contrast, for instance, Helheim site b (Sentinel-2 and Landsat 8 agree); Jakobshavn

site a (Sentinel-2 and Sentinel-1 agree); and Kangerlussuaq site a (Sentinel-1 and Landsat 8 agree). All three methods are orthorectified using different DEMs whose ages, accuracies, and resultant *Δh* values are variable. As such, it is likely that systematic biases are present in all three datasets and are spatially variable on a km-scale. However, the characteristics of the cross-track Sentinel-2 dataset will make it advantageous over Landsat 8-derived datasets in scenarios where a high density of observations is required, as the increased observation frequency will allow for a greater chance of successful image pairs over

critical dynamic periods, such as the melt season acceleration and late summer slowdown. In contrast, individual Landsat 8 velocity fields appear to be more precise, and may be preferable when the accuracy of individual velocity fields are necessary (e.g. comparing early and late season velocities). Sentinel-2 also provides a temporal advantage over Sentinel-1 datasets, which are limited to a fixed 6-day repeat cycle, and may also provide a valuable alternative for applying to small and steep glaciers, such as in high mountain regions, where the use of synthetic aperture radar for velocity extraction can be challenging (e.g.

Paul *et al.* 2021).

## 4.2 DEM origin

The differencing of orbital pair offsets from a reference flow field has been shown to be an effective method of reconstructing systematic orthorectification errors when the underlying DEM is not available. Furthermore, we show that orthorectification errors over Helheim Glacier are consistent with Sentinel-2 data being orthorectified using data sources present in the

Viewfinder Panorama 3" DEM (de Ferranti, 2014). This finding agrees with that of Kääb *et al.* (2016), who found similar agreement between a *Δh* estimated from the Viewfinder DEM and Sentinel-2 offsets for a test site in Northern Norway. Further testing is required to establish if the Viewfinder DEM and PlanetDEM continue to share high-latitude data sources, but if they





do then the freely available Viewfinder DEM opens new avenues for deriving high-latitude Sentinel-2 orthorectification error from analytical methods (Ressl & Pfeifer, 2018). To do so would require not only the Viewfinder DEM but a 'true' glacier

surface elevation DEM contemporaneous to image acquisition, which is difficult to acquire in rapidly-changing sectors of the cryosphere. Here, we highlight the utility of time-evolving DEMs and the potential of e.g. ArcticDEM strips (Porter *et al.* 2018) and monthly/annual composites thereof to address this challenge in the future.

### 4.3 Future datasets

The empirical offsets derived in this study are only applicable up to the 23rd August 2021. On this date (or the 30th March

2021 for coverage of Europe and Africa), L1C/L2A Sentinel-2 products switched to an improved geometric refinement with two major changes: (i) co-registration of scenes to a Global Reference Image (GRI); (ii) the use of a new DEM, the Copernicus DEM at 90 m resolution ("GLO-90"), for topographic correction. This new DEM is based on 2011 – 2015 radar satellite data acquired during the TanDEM-X mission, meaning there will be a discontinuity in $\Delta h$ between the pre- and post-August 2021 data.

There is no currently announced reprocessing of the Sentinel-2 archive with the new geometric refinements (cf., for example, Landsat Collection 2), so this discontinuity between pre- and post-2021 data will remain into the future. As such, the method outlined here remains applicable for the correction of Sentinel-2-derived glacier velocity across the wider cryosphere for Sentinel-2 L1C and L2A data between 2015 – 2021. The more recent dataset will reduce the value of $\Delta h$ and as such the orthorectification error in the new datasets - although as the DEM used for orthorectification consists of pre-2015 data, some

orthorectification error will remain. However, GLO-90 is available for public download (European Space Agency, 2021), allowing for correction via analytical methods (section 4.2).

### 5 Conclusion

By taking advantage of the complete Sentinel-2 record between 2017 – 2021, we demonstrate a method of correcting for systematic orthorectification error in glacier velocity fields derived from cross-track image pairs. This method is used to

produce a complete dataset of glacier velocity for four key marine-terminating glaciers across the Greenland Ice Sheet. Comparison with alternative datasets highlights the key advantages of the high temporal frequency of Sentinel-2 imagery, but this density of data also necessitates the use of statistical techniques to account for noise and uncertainty in the dataset. The transition to a new geometric refinement method for Sentinel-2 scenes beyond August 2021 will require new correction offsets to be generated, but the transition to a publicly available DEM (and the identification of potential 2015 – 2021 data sources in

this study) provides the opportunity for this to occur through analytical rather than empirical methods.

*Data availability.* The velocity datasets described here will be able to be accessed through the NSIDC upon final publication. The NSIDC also hosts access to the supplementary data used in this work: ITS_LIVE data (doi:10.5067/IMR9D3PEI28U),



MEaSUREs Sentinel-1 velocity data (doi:10.5067/6JKYGMOZQFYJ), GIMP DEM (doi:10.5067/H0KUYVF53Q8M), and

the GIMP mask (doi:10.5067/B8X58MQBFUPA). Version 1 of the 3" Viewfinder Panorama DEM for Greenland can be found

at http://viewfinderpanoramas.org/GL-ReadMe.html. Sentinel-2 scenes were accessed via the AWS Registry of Open Data

(https://registry.opendata.aws/sentinel-2). Velocity processing was performed using SETSM, available at

https://github.com/setsmdeveloper/SETSM.

*Author contributions.* TRC: conceptualisation, methodology, investigation, formal analysis, investigation, writing – original

draft, visualisation. IMH: conceptualisation, methodology, writing – review & editing, supervision, project administration,

funding acquisition. BNY: investigation, resources, data curation, writing – review and editing. MJN: investigation,

methodology, software, writing – review and editing.

*Competing interests.* The authors declare they have no conflict of interest.

*Acknowledgements.* This project was supported by grants 80NSSC18M0078 and 80NSSC18K1027 from the National

Aeronautics and Space Administration programme. The velocity fields contain modified Copernicus Sentinel data (2017–

2021), processed by ESA.

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
