# Peer review of "Empirical correction of systematic orthorectification error in Sentinel-2 velocity fields for Greenlandic outlet glaciers"

_The Cryosphere, 2022_

## Author Comment (AC1)

**THE OHIO STATE UNIVERSITY**

**Byrd Polar & Climate Research Center**

Glacier Dynamics Research Group

Scott Hall Room 123
1090 Carmack Road
Columbus, OH 43210

chudley.1@osu.edu

3 May 2022

Dear Dr Berthier,

**RE: Comments on tc-2022-33**

We have received 2 reviews on our manuscript tc-2022-33 ***Empirical correction of systematic orthorectification error in Sentinel-2 velocity fields for Greenlandic outlet glaciers***, both of which found our work methodologically sound and of relevance to the wider community. We are grateful to Drs Mouginot and Altena for their constructive comments.

We have adapted the manuscript and introduced new text and figures in light of the comments, which we describe in detail below. In addition, we have made minor textual changes to the manuscript that were not directly requested by reviewers, in order to improve the clarity of the manuscript in light of the reviewer recommendations, or to fix minor errors in the text. We have attached a revised manuscript with changes tracked. Unless otherwise specified, line numbers in our response to reviewers refer to change-tracked version of the manuscript.

Thank you for your consideration of our revised manuscript, which we hope is now acceptable for publication.

Sincerely,

Tom Chudley

On behalf of all co-authors

**Review 1: Jeremie Mouginot**

*Chudley et al. presents a study on the correction of the orthorectification error in surface ice velocity generated from Sentinel-2 acquisitions in Greenland. The results are sounds, clear and well described in the manuscript. Having a methodology to use the potential of Sentinel-2 images to track the dynamic evolution of Greenland's glaciers is a significant advance and will be useful for many other studies. The paper focuses mainly on the methodological aspects of remote sensing and as such could have its place in a journal more related to this type of subject. Nevertheless I believe that the topic will be of interest to the glaciological community and so could be published in The Cryosphere. Therefore, I recommend publication and have only minor comments below.*

We are grateful to Dr Mouginot for his helpful and supportive review. Below, we have outlined how we have revised the manuscript in light of his comments and recommendations.

**Comments:**

*An important point of the study to justify the processing Sentinel-2 images from different orbits (cross-track pairs) is the increase in the number of measurements that are useful for capturing rapid changes in velocity and producing dense time series. I think that the advantage of such an approach over considering only observations of similar orbits (repeat-track pairs) should be reinforced. In short, I think showing time series with and without the cross-track pairs could be useful to highlight the advantage over using repeat-track pairs.*

We have included a new figure (figure S4, see below) to address this comment and that of the minor comment addressing P12L250 by Reviewer #2. that separates cross-track and repeat-track pairs. Figure S[X]a shows this for all pairs in 2019 at Jakobshavn site a, whilst Figure S[X]b shows the same for only for short baselines (<10 days). This highlights the particular utility in creating dense time series consisting of short-term velocity pairs. We have highlighted this point and associated figure in the discussion at line 313-315.

[Figure]

*Figure S4: (a) Time-series of uncorrected observations at Jakobshavn site a, coloured by temporal baseline and with same-track pairs marked as crosses. (b) Same as a, but showing only temporal baselines less than ten days.*

*Similarly, the processing of Sentinel-2 data does not seem to provide additional information to that obtained by Sentinel-1 (Fig. 7). Would it be possible to find an example where S2 would fill a gap with respect to the already published time series in Sentinel-1 MEaSUREs (or Landsat-8 ITS_LIVE) ?*

We have updated figure S5 (figure S4 in previous draft) to additionally include Sentinel-1 MEaSUREs data showing GP regression applied to all datasets over the period of velocity slowdown at Store Glacier in 2018. This shows that the Sentinel-2 dataset is better able to resolve this period of rapid slowdown, which takes place over a period of ~2 weeks, than the comparative datasets, which cannot resolve the slowdown to less than one month. We have updated the text where we refer to fig. S5 (L298, L316) to better reflect this.

[Figure]

*Figure S5: Summer slowdown at Store Glacier site (a) in 2018, highlighting the inability to capture the rapid slowdown in sparse Landsat 8 and Sentinel-1 derived data. Points mark individual velocity measurements, solid lines are Gaussian Process models, and shaded regions mark the 2-sigma confidence interval.*

*Gaussian process regression could be further described. Although a short description of the kernels used is given, it seems difficult with the details provided to reproduce the results obtained in Figure 7.*

We have modified and further added to the description with the exact kernel functions used to aid the reader in replicating the processing (paragraph beginning L199). Although the explanation is short, it is not deceptively so: the scikit-learn library is both high-level and user-friendly. The core processing chain can be performed in six lines of code and an applicable example of the Rasmussen and Williams (20006) implementation for seasonally variable time-series datasets is outlined in a short tutorial [link] in the documentation.

*"We performed GP regression using the GaussianProcessRegressor() implementation in the Python scikit-learn library. We model ice velocity as the sum of two kernels (covariance functions) representing seasonal and short-term variability respectively. Our implementation follows Rasmussen and Williams (2006), ignoring long-term variability over the five-year period we assess. Seasonal variability is incorporated as the product of an exponential sine squared kernel (ExpSineSquared()) and a radial basis function kernel (RBF()). The exponential sine squared kernel has a fixed periodicity of one year and the length-scale as a free parameter. The radial basis function kernel has a free length-scale. We implement short-medium term variability as a rational quadratic kernel (RationalQuadratic()), with free length-scale and alpha (controlling the diffusivity of the length-scale) parameters. Finally, we incorporate velocity error estimates (section 2.1.3) into modelling directly via the alpha parameter of the GaussianProcessRegressor()."*

*In Fig.6 : The name "Jacobshavn" is not consistent with the text (Jakobshvan)*

This has been corrected in the new figure.

**Review 2: Bas Altena**

*The authors present an implementation/ workflow description, to use displacement products from different orbital tracks of Sentinel-2. They are not alone, as such efforts are becoming popular in other fields of remote sensing of the cryosphere as well e.g.: [Lavergne et al. 2021]. For this work the authors have opted for a more methodological approach, instead of a paper looking at the physical drivers, which fit into a journal such as TC. Seen in this light the work does not go deeper than a desciption, at several places the reader is left in the dark why certain steps are taken. The work would become more interesting when such heuristics and rationale are explained or appropriate earlier work is mentioned as implementation argument.*

We are grateful to Dr Alterna for his helpful and supportive review. Below, we have outlined how we have revised the manuscript in light of his comments and recommendations.

**Major Comments**

*Looking more detailed to the work, the advancement made by the authors is less clear to me. Though it is a valid implementation, through setting of a hypothesis and testing through emperical relations, it reads as an isolated peace of work. As such relations for Sentinel-2 (and Landsat 8) have already been set out by [Kääb et al. 2016] and [Altena & Kääb, 2017]. A brief description is given about a vector projection method as presented in [Altena & Kääb, 2017]. Though this is a stripped down version, for large scale processing pipelines, as demonstrated in this work. While the major work presented in [Altena & Kääb, 2017] deals with a framework of harmonization of different orbits and elevation offset. Which later is extended to changing topography over time [Altena et al. 2019]. The limitations of the projection method are correct (which is a stripped version), but these do not apply to the core framework. Hence, the presented work is halfway the work presented previously by others. If one assumes stable topography and a perfect co-registration (which is done by the authors), all cross-orbit displacements observing a region on the ground are related. Why are they then isolated? It does not look like an improvement...?*

We are grateful for this clarification of the relationship between this and the extensive work Dr Altena has done in the realm of Sentinel-2 correction. Our work follows on from the stripped-down method presented in Altena & Kääb (2017) in that it is simple and suitable for application to large-scale processing pipelines. In fact, as described in the text, our method *requires* a large-scale application in order to have enough data to reliably reconstruct the error. We have modified the text introducing Dr Altena's work in order to better reflect this (beginning L76):

*"Finally, recent work has begun to develop frameworks that harmonise orbit and elevation offsets in Sentinel-2 glacier velocity datasets that do not require any prior knowledge of Δh (Altena & Kääb, 2017; Altena et al. 2019). However, the complexity of these methods necessitate simplified pipelines for operational use and bulk processing (Altena & Kääb, 2017). The simplified method presented by Altena & Kääb (2017) takes advantage of the fact that, if glacier flow is known a priori..."*

Furthermore, we have rewritten the concluding sentence of this paragraph to better clarify the primary step forward we wished to make in our methodology (i.e. the limitation of the stripped-down projection method) (beginning L90):

*"…it would be desirable to develop of method that, like Altena and Kääb (2017), remains simple, computationally efficient, and does not require any prior knowledge of Δh, but is also able to produce a geographically continuous record of velocity that is not spatially limited by the relationship between satellite geometry and flow direction."*

*In general context is missing, as other efforts like presented in e.g.: [Rosenau et al. 2015] are not mentioned. Results of the change in temporal baselines is mentioned and a decision is taken, but an assessment is missing, while [Millan et al. 2019] do show interesting results on this aspect. Hence, why is such knowledge not taken into account, and do the authors branch off?*

We are grateful for the identification of highly relevant papers that were not included in our original submission. We have now discussed Rosenau et al. (2015), and their own approach to addressing orthorectification error, in the introduction (beginning L65):

*Rosenau et al. (2015) provide their own improved orthorectification for Landsat imagery by orthorectifying L1G data to the ~1" ASTER Global Digital Elevation Model (GDEM) V2, rather than the ~30" Global 30 Arc Second Elevation (GLOTOPO30) dataset that was standard for L1C products at the time. However, the non-orthorectified product (L1B) for Sentinel-2 is not made available, meaning that this approach is not viable.*

Please see our responses below to line comments P8L176 and P9L205 for a more detailed address of the temporal baseline and the relevance of the work of Millan et al. 2019.

*As a final note, this review needs to be done on paper, but if done in speech, it would have been on a friendly tone. I do think the authors present nice work, and this work should be seen as work in progress, like any scientific enedavour. My comments are encouragements, with the aim to set this work to a higher level. A potential is present within this effort, but not exploited to its fullest.*

We are grateful to Dr Altena for his helpful and thorough review and of course take his comments in the spirit with which they were intended.

**Minor Comments**

*- most people scan a paper by reading the abstract first, but context is skewed here. Why would people use cross-track data?*

We have added an additional sentence to the abstract (L11-13), highlighting the motivated for cross-track image pairs:

*"As a result, most standard processing chains ignore cross-track pairs, which limits the opportunity to fully benefit from Sentinel-2's high-frequency observations during periods of intermittent coverage or for rapid dynamic events."*

*Typically, this is an exotic way of processing, thus why the authors put much emphasis on stating the presence of enormous errors in ESA products is a bit strange. It misses the overview, as this work is a nice contribution, but has limited impact.*

The consideration of the errors introduced by PlanetDEM-90 is, to us, one of the key conclusions of the paper. Major effort has been put forward, indeed primarily by Dr Alterna, into developing methods of correcting errors introduced by this DEM into cross-track velocity pairs from Sentinel-2. All of these methods, in some way, work around the fact that the DEM is not publicly available. Here, we show strong evidence that the underlying DEM for Greenland is in fact available in an alternative public dataset, opening up a clear avenue to work towards analytical solutions to correct orthorectification errors in the 2016-2021 velocity data. As ESA has not announced plans for a reprocessing of the 2016-2021 Sentinel-2 data to the updated Copernicus DEM in a way comparable to the Landsat Collection-1/2 products, this is a key finding in providing these analytical solutions for this period in the future.

In response to this and a comment P17L346 below, we have updated the conclusion (beginning L369) to better reflect the key parallel impacts we consider our work to make: (i) a simple and efficient empirical method for correcting Sentinel-2 glacier velocity fields in large-scale datasets; and (ii) advances in determining the underlying DEM sources for 2015-2021 Sentinel-2 imagery over Greenland, and the opportunities this raises for analytical solutions.

*- Gaussian Processes are popularized in our field by [Huggonet et al. 2021], hence this is an argument to use this approach. Now justification is missing.*

Thank you for identifying this – we have further referenced Huggonet et al. in the methods section (L193).

*- Why are these specific glaciers used, if cross-orbits are of interest, Northern Greenland is very interesting, since overpasses amost occur every day, see Fig.1 [Altena et al. 2019].*

The study glaciers we have chosen are well-studied outlet glaciers that are significant in terms of contribution to GrIS discharge, are of ongoing interest to the glaciological community, and represent a diverse range of seasonal behaviours. Interestingly, we chose glaciers outside Northern Greenland for precisely the reason mentioned here: Northern Greenland glaciers already have a dense temporal coverage even without cross-track pairs. Focussing on glaciers further south allows us to examine the viability of filling in coverage with cross-track pairs even where coverage is non-optimal.

**Typos and Details**

*Since I am not a native speaker, I am not able to give any feedback on typos, nonetheless some details might be improved:*

*p1 l29 Landsat8 is now a fleet together with Landsat9, having same orbit repeat cycle of 8 days*

Fixed – added "*now 8 days from 2022 onwards with the addition of Landsat 9*" (L31-32)

*p1 l29 Sentinel-2A and Sentinel-2B, need capital letters*

Fixed (L31)

*p2 l41 it might be a cultural thing, but please do use words instead of newly introduced abbreviations, especially as this is abbreviation is later changed into something else, and not used anymore. If you like abbreviations, than help the reader a bit and include an appendix with a nomenclature.*

We have removed the 'relative orbit' abbreviation from the paper (L43, and throughout).

*p2 l56 with the advent of so many same orbit repeat aquisitions, orthoretification errors are not a "significant" issue. The cross-orbit velocities are a "nice-to-have". Please do not oversell.*

The use of 'significant here refers to the scale of the DEM errors relative to Landsat 8, rather than a reflection on the relative importance of the problem. We have rephrased to be more specific: *"large vertical DEM errors (10s-100s metres)"* (L56)

*p3 l67 "is not freely available" -> "is a commercial product"*

Fixed (L73)

*p3 l81 such a methodology is presented in the [Altena and Kääb] study, so what is the unique contribution...?*

We have rewritten this section (L90-93) to better highlight the distinct features of our method relative to the Altena & Kääb study, as discussed in our response to the first major comment.

*p4 l93 please consider using a logarithmic colorbar, as most (~95%) of the figures are dark blue now...*

The use of a logarithmic scale for velocities would indeed by better at highlighting variation at the low end of velocities but it also acts to reduce the colour scale at the upper end, which is where the key variations detectable by this method exist (in particular, it would dampen the visible variation in figure 5). In the interests of keeping the colour scale consistent throughout the paper, we have chosen to retain a linear colour scale.

*p6 l117 AOI, why is this acronym introduced, if it is only used here?*

We retain this acronym as (i) it is commonly used across the literature; (ii) although limited in scope, we use it a total of six times in a relatively short period, which would be unwieldy to read otherwise; (iii) Our use of the phrase in the figure 2 flowchart, where "Area of Interest" would be hard to fit in to the box.

We have, however, noticed that mistakenly introduced the phrase twice in the first submission. We have fixed this, and now only introduce it once on L121.

*p6 l130 here the newly introduced RO acronym is again replaced, what is the use?*

We have now removed the RO acronym (see above). Cross-track and repeat-track are not replacing the 'relative orbit' term directly: instead, they refer to configurations of relative orbits for the respective velocity pairs. One solution to maintain an internal consistency may be to refer to these as *cross-relative orbit* and *repeat-relative orbit* respectively, but this is unwieldy. In referring to relative orbits, we are staying consistent to ESA/Copernicus programme terminology; in using cross-track and repeat-track, we are reverting to phrases commonly used in the literature. We think this is a suitable compromise.

*p6 l130 the orbital number is even in the filename, as is also the case for Landsat, so this line is obsolete.*

For simplicity, we have changed this sentence to "*Velocity fields are grouped by the relative orbits of their respective source image pairs*" (L142)

*p7 l145 where does this 5x heuristic threshold comes from?*

This threshold was identified based on manual assessment of a range of options. Below 5 pairs, the quality of the correction fields is notably degraded (as we are calculating the median from a low number of observations, this is perhaps unsurprising). We now better highlight this when the concept is first introduced (L150-251).

*P7 l148 why does the assumption of stable geometry still hold, even for an highly dynamic outlet as Sermeq Kujalleq (Jakobshavn Isbræ) [Joughin et al. 2020 & Riel et al. 2021]?*

Our justification of this assumption is outlined in the paragraph beginning L154: ultimately, the initial value of Δh is so large that, even using worst-case figures for a highly dynamic glacier, the maximum displacement error contributed by this assumption over our study period is at most on the order of ~1 m and thus subsumed by other error terms.

However, this does raise the point that this assumption will not hold for the new Copernicus DEM, which will have lower initial Δh values. We have added new discussion of this in section 4.3 (L361-363):

"*Additionally, the lower initial Δh value will likely invalidate the assumption we make in this study that further elevation change will have a negligible impact on our assumption of stable geometry.*"

*p7 l157 the description of vertical DEM error is to broad, since it is not random but has mostly a systematic effect.*

We have rewritten this sentence in order to better communicate our intent, which is that our correction scheme is designed to correct for surface elevation change which we assume is largely negligible outside of the ice sheet. We update the sentence to read as such (L169-171):

"*Displacements are corrected only over ice […], as our correction scheme is designed to correct for surface elevation change which we assume is largely negligible outside of the ice sheet boundaries.*"

*p7 l162 it is not clear to me if stable ground is used, to co-register the imagery? please write it down, or give a motivation why not*

We have made this further explicit in our methods section (L136-137):

"*…and applied the relative sensor model bias compensation module as co-registrations. For co-registration, we define the off-ice region using the Greenland Ice Mapping Project (GIMP) ice mask*"

*p8 l166 why are velocities used here, while for orthorectification the temporal baseline is almost irrelevant. Please use appropriate units, i.e.: meter.*

Error in absolute displacement should be broadly consistent across all velocity fields. However, the same displacement has a much large impact on shorter timescales, and hence

error will be strongly dependent on the temporal baseline (see e.g. Figure [S3] – and also Millan et al. 2019 Fig. 4). We discuss this in the paper section 3.2 (L272-274). Ultimately, it is velocity error that is important to consider when differencing and filtering the fields, and also an inputs to our Gaussian processing pipeline. Reporting errors in velocity in further consistent with previous work (e.g. Millan et al. 2019).

*p8 l169 this is a strange formulation, but why is this classical statistic used and not robust measures like "median of absolute difference" (MAD)? Also, the registration of Sentinel-2 has a flight line dependent positioning error (see fig.15 in [Kääb et al. 2016]), hence treating U and V as uncorrelated delutes the effectiveness of this threshold (see fig.2 in [Altena et al. 2021]).*

We use standard deviation to be consistent with our comparative products (in particular, ITS_LIVE). The use of standard deviation is also consistent with previous work (e.g. Millan et al. 2019). The method outlined in Altena et al. 2021 provides an interesting perspective on calculating an isotropic uncertainty, and we look forward to begin implementing it in future pipelines upon the publication of the final work.

*p8 l174 "errors" > "deviations/differences"*

Clarified to RMSE error estimate (L188)

*p8 l176 please justify your decision, and have a look at [Millan et al. 2019]*

We mistakenly referred to Figure S1 here and not Figure S3, which shows the clear outlier of 2-day fields vs 3+ to properly justify our decision – we have corrected this in the text (L190). We think that the recommendation to consult Millan et al. 2019 is referring us specifically to section 3.3.6, which suggests that a good rule of thumb for detecting velocity changes is that $2\sigma$ must be <10% of the flow speed ($\sigma$ is calculated as the standard deviation in ice-free areas). Our errors of ~1 m/day (Fig S3) suggest that 3-day+ fields are appropriate for identifying change at rapidly regions of fast-flowing outlet glaciers. Please note also that our use of Gaussian Processing allows for further management of uncertainty in the stacked time series, allowing useful information to be contributed even from fields with higher uncertainties.

*p8 l180 "estimate" > "distill" and what is "true velocity", and are you able to back this claim, please rephrase*

We retain estimate but replace 'true' with 'continuous' (L193).

*p9 l205 please see [Millan et al. 2019]*

We think that this comment is suggesting that we could/should use >30 day baselines in order to be able to better detect change over slower-flowing sectors of our study AOIs. Whilst this is true, we do not do so for three reasons: (i) Our focus here is to show the validity of our method in detecting rapid variation (~days-weeks) in faster-flowing outlets, (ii) long baselines (on the order of hundreds of days) may not ultimately be useful in extracting useful information about the seasonal dynamics of the Greenland Ice Sheet, and multidecadal change has already been assessed (e.g. Tedstone et al. 2015); and (iii) Processing the pairs is time- and resource-intensive and, based on our focus outlined in points (i) and (ii), there are ultimately practical decisions to make as to how many pairs is desirable to process. Our

focus here is on extracting a dense high-temporal-resolution dataset, so for us the thirty day maximum baseline suits our needs.

*P9 l207 why are velocity units used as a threshold for variable temporal base line data?*

See our response above to P8L166.

*p9 l210 please indicate the flight direction of the Sentinel-2 satellite, similar to inSAR maps.*

We provide this information within Figure S2. We consider it more useful as a separate map as we can also provide information about the location of the orbit relative to the AOI. We now include a reference to Figure S2 in the caption of Figure 3 (L226)

*p10 l223 please give an indication of the intersection angle, how much is the base-to-height or its angle*

External imaging geomety could be useful in determining terrain-induced errors. However, since we are restricted to using orthorectified imagery provided in the tiling scheme, imaging geometries such as base to height ratio are less useful for determining errors. Indeed, one of the main objectives of the presented method is to derive an empirical terrain correction that is scaleable to large datasets and independent of specific pair geometries.

*p10 l228 why so much hypothesis, while cross-track photogrammetry is around for some time?*

We remove 'hypothetically' from this sentence, which implied that it was speculation rather than an expected result (L243)

*p11 l231 "true" > "reference"?*

Fixed (L246)

*p12 l250 the figure can be improved, mayby make most stuff black and let the red points change in colour for different time intervals? As Sermeq Kujalleq (Jakobshavn Isbræ) seems to have several consistent off-sets. Where do they come from, please describ it in the text, this makes it more informative to the reader. Is it also interesting to look at the flow direction? So assumptions on this aspect might propagate into this variance? Maybe scatter plots are not the best to use, thus connsider using line plots to draw the temporal baselines as e.g.: fig.4 in [Charrier et al. 2022]*

Thank you for providing feedback to improve this figure. Previously, including colour scale information on the points in the main figure created a visually noisy and difficult to interpret dataset. We have instead included a new figure (figure S4, see below) to address both this comment and that of the first comment of Reviewer #1. In it, we visualise the Jakobshavn data coloured by the temporal baseline of the scene, showing that the consistent offsets relate to the temporal baseline of the velocity pairs, as absolute displacement errors will have a higher relative effect on short-baseline fields. We note this and include a reference to the figure at L273 in Section 3.2 of the text.

[Figure]

*Figure S4: (a) Time-series of uncorrected observations at Jakobshavn site a, coloured by temporal baseline and with same-track pairs marked as crosses. (b) Same as a, but showing only temporal baselines less than ten days.*

*p12 l254 why not compare against off-glacier stable terrain?*

Our purpose here is to compare our derived data against comparative time-series datasets. Our comparison against the uncertainty comes in the form of our error estimates.

*p12 l261 this method does not resolve this issue of dynamic thinning, so why so specific about the cause?*

Here we simply intended to refer to the total Δh, which, in these contexts, largely occurs from dynamic thinning. We agree that this unclear, and have removed the reference to dynamic thinning from the text (L276).

*p13 l264 "high uncertainty of optical feature tracking", where does this loose claim come from?*

We agree with this assessment and have removed it from the final text (L279)

*p13 l265 please rephrase the GP sentence*

In combination with comments above, we have removed 'true' in favour of 'continuous' (L280).

*P13 l271 this is interesting, why is that? Please give more depth to the subject*

We agree that this is an interesting feature. We address this in the Discussion section in paragraph beginning L323. The cause is unclear, but we consider it likely related to underlying vertical uncertainty in the Sentinel-1 correction DEM at the specific AOI we sample here.

*p17 l346 very vague conclussion, but this can be improved if more in depth analysis are done.*

Based on this and above comments, we have modified the conclusion to better highlight the two parallel contributions we consider out study to make: (i) a simple and efficient empirical method for correcting Sentinel-2 glacier velocity fields in large-scale datasets; and (ii) advances in determining the underlying DEM sources for 2015-2021 Sentinel-2 imagery over Greenland, and the opportunities this raises for analytical solutions (beginning L369).

[revised manuscript text omitted]